

# Analysis of drought and heat stress response genes in rice using co-expression network and differentially expressed gene analyses

Gaohui Cao[1,*], Hao Huang[1,*], Yuejiao Yang[1], Bin Xie[2] and Lulu Tang[1]

[1] Department of Cell Biology, School of Life Sciences, Central South University, Changsha, Hunan, China
[2] State Key Laboratory of Hybrid Rice, Wuhan University, Wuhan City, Hubei Province, China
[*] These authors contributed equally to this work.

## ABSTRACT

Studies on *Oryza sativa* (rice) are crucial for improving agricultural productivity and ensuring global sustenance security, especially considering the increasing drought and heat stress caused by extreme climate change. Currently, the genes and mechanisms underlying drought and heat resistance in rice are not fully understood, and the scope for enhancing the development of new strains remains considerable. To accurately identify the key genes related to drought and heat stress responses in rice, multiple datasets from the Gene Expression Omnibus (GEO) database were integrated in this study. A co-expression network was constructed using a Weighted Correlation Network Analysis (WGCNA) algorithm. We further distinguished the core network and intersected it with differentially expressed genes and multiple expression datasets for screening. Differences in gene expression levels were verified using quantitative real-time polymerase chain reaction (PCR). *OsDjC53, MBF1C, BAG6, HSP23.2*, and *HSP21.9* were found to be associated with the heat stress response, and it is also possible that *UGT83A1* and *OsCPn60a1*, although not directly related, are affected by drought stress. This study offers significant insights into the molecular mechanisms underlying stress responses in rice, which could promote the development of stress-tolerant rice breeds.

## INTRODUCTION

Rice (*Oryza sativa*) is a vital cereal that is extensively grown worldwide, serving as a fundamental source of sustenance for approximately 50% of the global population (*Ashkani et al., 2015*; *Li et al., 2018*). In recent decades, with the rapid increase in the total global population and the demand for sustenance security, the importance of rice genetics and breeding has become particularly critical particularly critical (*Huang et al., 2016*). Studies focusing on the generation of advanced rice genotypes aim to improve the yield, quality, and resilience of rice plants to biotic and abiotic stressors, such as pests, diseases, salt, drought, and heat (*Raj & Nadarajah, 2022*; *Vo et al., 2021*). With the deterioration of the global climate, the frequency and severity of drought and heat waves are expected to increase in

Corresponding author
Lulu Tang, lltang@csu.edu.cn

many rice-growing regions (*Saud et al., 2022*; *Zandalinas, Fritschi & Mittler, 2021*). Rice is an aquatic plant predominantly grown in lowland areas that are often subjected to flooding, making the crop more vulnerable to drought and heat stress (*Jagadish, Murty & Quick, 2015*; *Ji et al., 2012*; *Sahebi et al., 2018*).

Advances in rice breeding and biotechnology and genetic cultivar improvement have played a significant role in increasing the drought resistance of rice while enhancing its ability to adapt to hot environments (*Shen et al., 2022*; *Wang & Han, 2022*). Research into the molecular mechanisms underlying drought and heat adaptability in rice can facilitate the creation of novel rice cultivars with improved stress tolerance (*Kim et al., 2020*; *Liu et al., 2020*).

Several key genes that confer drought tolerance in rice have been identified. *UGT85E1*- and *OsWRKY5*-mediated enhancement of the abscisic acid response has been shown to improve drought stress tolerance (*Lim et al., 2022*; *Liu et al., 2021*). *OsNAR2.1* plays a fundamental role in nitrate absorption and translocation; thus, its expression level is positively correlated with drought resistance in rice (*Chen et al., 2019*). *OsRINGzf1* regulates aquaporins during drought stress (*Chen et al., 2022*). The expression levels of photosynthesis-related genes, such as *CA1*, also change under drought stress (*Auler et al., 2021*; *Li et al., 2020*). The overexpression of *Arabidopsis UBC32* improves drought tolerance in rice (*Chen et al., 2021*). These genes are involved in various processes such as hormone signaling pathways, osmotic regulation, and photosynthesis.

*OsRab7*-mediated modulation of osmolytes, antioxidants, and genes that respond to abiotic stress can lead to improved grain yield and enhanced ability to withstand heat in transgenic rice (*El-Esawi & Alayafi, 2019*). *OsTT1* plays a protective role against heat stress by eliminating denatured proteins that are cytotoxic and preserving thermal response processes in cells (*Li et al., 2015*); *OsNTL3* and *OsbZIP74* have a similar mechanism (*Liu et al., 2020*). *HES1* maintains the stability of the photosynthetic system under high-temperature stress (*Xia et al., 2022*). These genes are associated with heat shock proteins (HSPs), antioxidant enzymes, protein synthesis, and photosynthesis.

In summary, research on drought and heat durability in rice is critical for ensuring global food security, adapting to extreme climate change, and improving agricultural productivity (*Tyczewska et al., 2018*). Previous studies have provided valuable insights into the physiological and molecular aspects of stress responses in rice (*Lakshmanan et al., 2016*). However, one significant gap limiting the current literature is the incomplete identification and understanding of the key genes and regulatory networks involved in drought and heat stress responses in rice. Although many stress-responsive genes have been identified, they represent only a small fraction of the vast number of genes in rice. Existing studies are unable to compare the significance of these genes in stress responses. This limits our ability to develop targeted strategies for enhancing stress tolerance in rice varieties.

To comprehensively analyze the molecular mechanisms underlying drought and heat responses in rice, a set of RNA-seq data from the Gene Expression Omnibus (GEO) database was selected, which contained different gradients of drought and heat treatments, and the data were compared with that in multiple datasets that were subjected to either drought

or heat stress. The integration of diverse datasets and the utilization of advanced analytical techniques allowed us to overcome the limitations of individual studies and provide a more holistic view of the molecular mechanisms underlying stress responses in rice. The present study enhances our understanding of the molecular mechanisms underlying drought and heat stress adaptation in rice and can be useful in discovering new and more important genes that could serve as candidates for genetic breeding purposes. Portions of this text were previously published as part of a preprint (https://doi.org/10.21203/rs.3.rs-3047406/v1).

## MATERIAL AND METHODS

### Data collection

Multiple gene expression profiling datasets, including high-throughput sequencing (Illumina HiSeq 2000/Illumina HiSeq 4000/Illumina NovaSeq 6000) and array datasets (Affymetrix Rice Genome Array Platforms), were sought and retrieved from the GEO database (https://www.ncbi.nlm.nih.gov/geo/). These high-highthroughput sequencing datasets included GSE221542, GSE168650 (*Kan et al., 2022*), and GSE159816 (*Zu et al., 2021*). These array datasets included GSE136746 (*Ps et al., 2017*), GSE41648 (*Sharma et al., 2021*), GSE14275 (*Hu, Hu & Han, 2009*), GSE93917 (*Wang et al., 2020*), and GSE83378 (*Wei et al., 2017*) (Table 1). Gene symbols for these GEO datasets were annotated using the National Center for Biotechnology Information (NCBI), Rice Annotation Project database (RAP-db) (https://rapdb.dna.affrc.go.jp/), and the Rice Genome Annotation Project (http://rice.uga.edu/index.shtml). The data were processed using R (version 4.2.3) and RStudio (version 2023.03.0) software. GSE221542 contains 15 samples, including three water levels and two heat levels, each with three replicates.

### Weighted gene co-expression network analysis of drought/heat response genes

Counts per million were computed to standardize the sequencing depth of RNA-seq data using the R package "edgeR" (*Robinson, McCarthy & Smyth, 2010*). Using the weighted gene co-expression network analysis (WGCNA) (*Langfelder & Horvath, 2008*) package in R (version 4.2.3), aco-expression network was constructed using the following steps. First, the average expression of each gene under different levels of drought or heat stress was calculated, and genes that did not exhibit any changes in expression were filtered out. Second, normalization of gene expression levels to a range of 0–1 was followed by the calculation of Pearson's correlation coefficients, which is used to measure the similarity of co-expression between genes. Third, to ensure a scale-free network distribution, an appropriate beta value was selected for the adjacency matrix weights to construct a topological overlap matrix for module clustering and segmentation. Finally, to select modules related to drought or heat responses, the relationship between each network module and the sample phenotype was analyzed.

Gene Ontology (GO) terms were used to enrich selected genes (*Tian et al., 2017*). The analysis results were presented using the R package "clusterProfiler" for visualization (*Yu et al., 2012*). Kyoto Encyclopedia of Genes and Genomes (KEGG) enrichment (*Kanehisa & Goto, 2000*) analysis was also performed using the R package "clusterProfiler" (*Yu et*
**Table 1   Raw data information from GEO.**

| Name | Dataset | Cultivar | Tissue | Samples |
|---|---|---|---|---|
| GSE221542 | GSE221542 | Nipponbare | whole shoot | all |
| GSE168650X | GSE168650 | NIL-TT2HJX | developing aerial tissues | heat *vs* control |
| GSE168650-32 | GSE168650 | NIL-TT2HPS32 | developing aerial tissues | heat *vs* control |
| GSE136746-N22 | GSE136746 | Nagina22 | panicle | heat *vs* control |
| GSE41648-Ann | GSE41648 | Annapurna | seedling | heat *vs* control |
| GSE14275 | GSE14275 | ZhongHua 11 | seedling | heat *vs* control |
| GSE159816-WT | GSE159816 | wild type | leaf | drought *vs* control |
| GSE159816-idr11 | GSE159816 | idr1-1 | leaf | drought *vs* control |
| GSE93917-nadk1 | GSE93917 | osnadk1 | leaf | drought *vs* control |
| GSE93917-WT | GSE93917 | wild type | leaf | drought *vs* control |
| GSE83378-MILT | GSE83378 | MILT1444 | panicle | drought *vs* control |

*al., 2012*). Using the CytoHubba (*Chin et al., 2014*) plugin of Cytoscape (3.9.1), based on the shortest paths, every gene of the key module was scored using the Maximal Clique Centrality (MCC) method, and the top 20 hub genes were selected.

## Differentially expressed gene analysis with DESeq2 and GO enrichment in R

Differentially expressed gene (DEG) analysis was performed using the R package DESeq2 (*Love, Huber & Anders, 2014*). Raw count data from the RNA-seq experiments were imported into R, and genes with low expression were filtered using the "filterByExpr" function. Next, the "DESeqDataSetFromMatrix" function was used to create a DESeq2 object, which was then used to estimate size factors and dispersions using the "estimateSizeFactors" and "estimateDispersions" functions, respectively. A false discovery rate cutoff of 0.05 was applied to identify genes that were significantly differentially expressed, based on an absolute log2 fold change $\geq 1$ and an adjusted $p$-value $\leq 0.05$. All data analyses were performed using R software (version 4.2.3).

GO enrichment analysis was also performed to analyze DEGs using a previously described approach (*Tian et al., 2017*; *Yu et al., 2012*).

## Intersection of hub genes and DEGs for candidate key genes

The top 20 hub genes from the filtered key modules were compared with the DEGs obtained from the filtering process. Based on their intersection, the candidate key genes along with their log2 fold change values were obtained. The Rice Gene Index (RGI) (https://riceome.hzau.edu.cn/) was used to determine the gene ID corresponding to the rice gene (*Yu et al., 2023*).

We searched for datasets on drought or heat treatments in the GEO database (Table 1). Count data were processed using the same method as above but not filtered for log2 fold change $\geq 1$ and $p$-value $\leq 0.05$. For array data, online GEO2R analysis was performed, and a matrix table containing the log2 fold change, $p$-value, and adjusted $p$-value data was downloaded.

Using the "pheatmap" package in R (version 4.2.3), the log2 fold change calculated from the different array or count data treatments was clustered and plotted. Key genes with high and stable expression levels were selected for further experiments.

## Plant materials

The model rice variety *Oryza sativa* Nipponbare was subjected to appropriate environmental conditions, drought stress, and heat stimulation, as well as RNA extraction for quantitative real-time polymerase chain reaction (qRT-PCR).

Nipponbare rice seeds were germinated for 3 days at 30 °C. cultivated in Yoshida Rice Medium (Coolaber, Beijing, China) for 10 days (12 h light at 30 °C and 12 h dark at 27 °C every day). The control group was directly sampled. In the drought treatment group, the samples were grown for 10 days in the medium containing 2 g/L mannitol. In the heat treatment group, on the 10th day, rice plants grownwhich in normal medium werewas exposed to a temperature of 40 °C for 1 h and then sampled. The stress modeling method reference was based on the GSE221542 dataset.

## RNA extraction

Whole shoot tissues (100 mg) from different treatment groups were weighed and placed in a grinding tube containing steel beads. The grinding tubes were immersed in liquid nitrogen for 10–20 min. Finally, the samples were freeze-ground at −20 °C for 120 s and returned to liquid nitrogen for storage. RNA extraction was performed using the FastPure Universal Plant Total RNA Isolation Kit (Vazyme, Nanjing, China), and the extracted total RNA was stored at −80 °C.

## Quantitative real-time PCR

cDNA was synthesized using the Revert Aid First Strand cDNA Synthesis Kit (Thermo Scientific, Watham, MA, USA). qRT-PCR analysis was performed using a LightCycler 96 (Roche, Basel, Switzerland). *eEF1* was used as the reference gene (*Ambavaram & Pereira, 2011*). Gene sequences were searched using Phytozome (https://phytozome-next.jgi.doe.gov/), and qRT-PCR primer sequences were designed using the primer blast tool of NCBI (https://blast.ncbi.nlm.nih.gov/Blast.cgi). The primers used in this study are listed in Table 2. See Supplementary 1 for the MIQE checklist (*Bustin et al., 2009*).

The relative expression level of target genes was calculated based on the $2^{-\Delta\Delta Ct}$ method for normalization (*Livak & Schmittgen, 2001*). The normalized qRT-PCR data were analyzed using the $t$-test to determine statistically significant differences in gene expression between the control and experimental groups (*Wilson & Worcester, 1942*). Statistical significance was set at $p < 0.05$.

## RESULTS

### Construction of co-expression network

The workflow followed in this study is depicted in Fig. 1.

WGCNA was applied to analyze the GSE221542 dataset, with a scale-free topology model fitting degree of 0.8 and a soft threshold of 30 selected for network construction

**Table 2  Primer information for qRT-PCR.**

| Gene name | MSU-ID | Group | 5′ primer | 3′ primer |
|---|---|---|---|---|
| eEF1 | LOC_Os03g08010 | Reference | GATGATCTGCTGCTGCAACAAG | GGGAATCTTGTCAGGGTTGTAG |
| BAG6 | LOC_Os02g15930 | Green | GTTGAAAGTAGTGTGTCAGCT | AAGGATACTGATGAGTCCCC |
| HSP23.2 | LOC_Os04g36750 | Green | GGTGGAGGTGGAGGACAA | CCAGAACCTGCCGTAGGA |
| OsDjC53 | LOC_Os06g09560 | Green | GATTTCCTCGGCGAGATGG | ACGAACAGCTGCTGCAA |
| MBF1C | LOC_Os06g39240 | Green | AGGTTGAGCGGCAACATC | CGCATCGCCTGGTTCAC |
| HSP21.9 | LOC_Os11g13980 | Green | CGTACGGCTACGGCTACAT | TCCTTCCAGTCGCACCTC |
| UGT83A1 | LOC_Os03g55030 | Darkmagenta | GGCGTCCTCAACGAGAAG | CAGACGAGGTCGAAGATGATG |
| OsCPn60a1 | LOC_Os12g17910 | Darkmagenta | CAAGGCTGTCCTTCAGGATATT | TGTCCCAAGTTGCTCTTCAG |

(Figs. 2A–2B). A hierarchical clustering process was used to create a tree-like structure representing genes. Subsequently, gene modules were determined using the dynamic cutting method, followed by the calculation of the eigenvector value of each module. Similar modules were then merged to identify distinct modules, which were assigned different colors for better visualization (Fig. 2C).

## Co-expression network module analysis

Six modules, namely, black (1,550 genes), green (1,646 genes), dark orange (3,658 genes), dark magenta (535 genes), royal blue (9,432 genes), and gray (234 genes), were obtained. The modules showed either positive or negative correlation with drought or heat stress, and the genes within these modules were either upregulated or downregulated, suggesting that the genes respond differently under different stress conditions. The green module with heat and the dark magenta module with drought had the highest positive correlation coefficients (0.98 and 0.71, respectively) (Fig. 3A). According to the scatter plots, the genes in the green module were highly correlated with heat stimulation, whereas the genes in the dark magenta module showed a weak association with drought stress (Figs. 3B–3E). The other modules showed low correlation with heat or drought stress (Fig. S1).

Cytoscape software was used to process the dark magenta and green modules separately and visualize the co-expression network obtained from WGCNA. Genes in the network were scored using the maximal clique centrality (MCC) method, and the top 20 hub genes with the highest correlations with other genes were selected (Figs. 3F–3G). These genes were located at the most central positions in the co-expression network, and they may play a central regulatory role in drought and heat stress.

In addition, KEGG enrichment analysis showed that genes in the green module are involved in processes such as protein synthesis in the endoplasmic reticulum and RNA splicing, whereas genes in the dark magenta module are involved in essential processes such as carbon metabolism, synthesis of amino acids and coenzyme factors, and glycerolipid metabolism. In addition, both gene modules are involved in carbon fixation in photosynthetic organisms (Fig. 4A). The results of the GO enrichment analysis showed that genes in the green module were related to biological processes such as cellular response to stimuli, phosphorylation, and signaling, whereas genes in the dark magenta module were involved in phosphorus metabolism and phosphate-containing compound

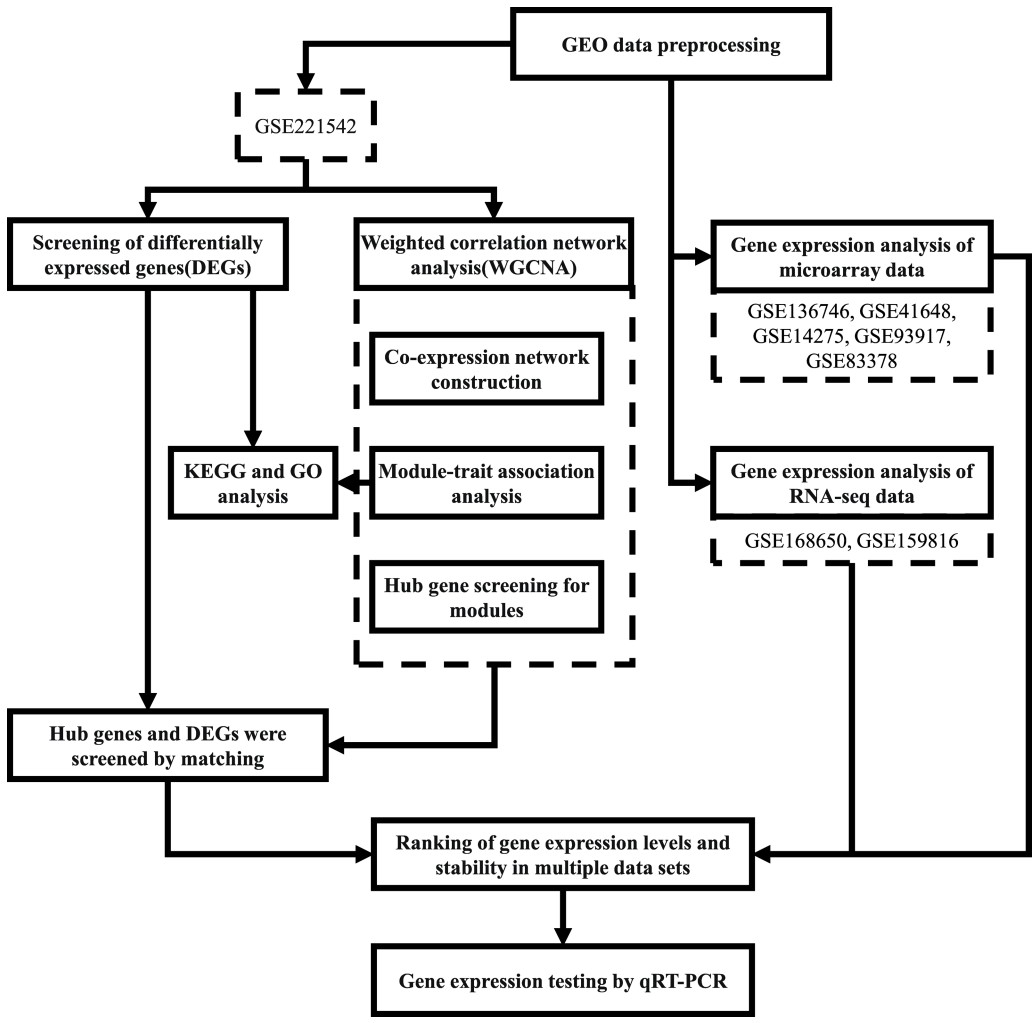

**Figure 1** **Workflow of the present study.**

metabolic processes. Although genes of both modules are expressed in the cytoplasm and vesicles, genes of the green module are expressed in membrane-bound organelles only. In the green module, the expressed proteins exhibited transferase activity and nucleic acid binding, whereas those in the dark magenta module exhibited catalytic activity and metal ion binding (Fig. 4B).

Construction of the gene co-expression network narrowed the range of candidate genes, and the 20 hub genes obtained by screening made the follow-up study more convenient.

## DEG analysis

Differential gene analysis was performed on two modules of the GSE221542 dataset: severe drought stress and control and prolonged heat stress and control (Fig. 5A). The stress group with severe drought and long-term heat shock was selected for analysis (Control: GSM6883305-7, Drought: GSM6883299-301, Heat: GSM6883311-3). We found 484 and 1,559 had increased and decreased expression levels, respectively, in the drought stress

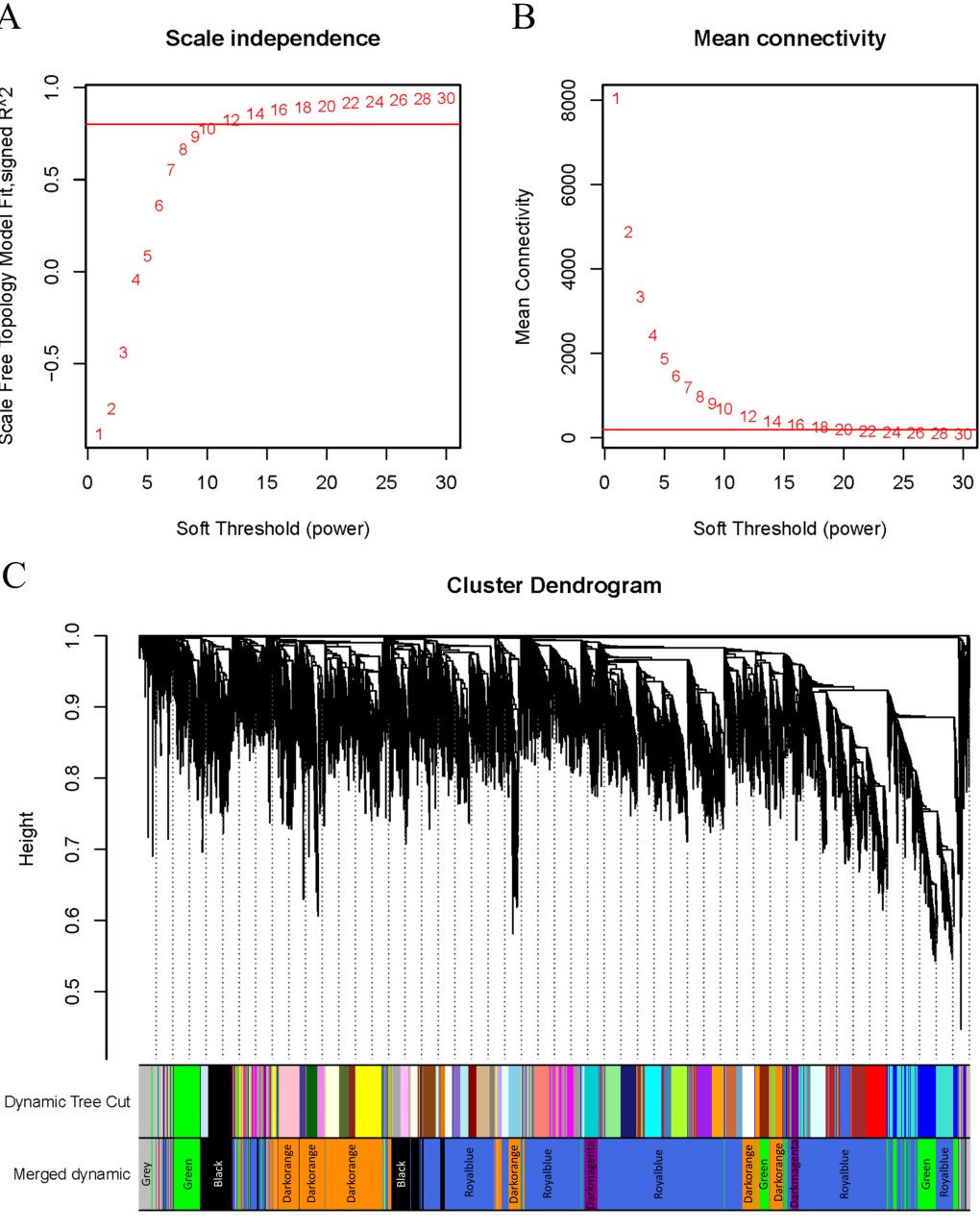

**Figure 2 $\beta$ setting and clustering for WGCNA.** (A, B) Network topology for different soft thresholding powers. The $x$-axis represents the weight parameter $\beta$. The $y$-axis in panel (A) represents the square of the correlation coefficient between log(k) and log(p(k)) in the corresponding network. The $y$-axis in panel (B) represents the mean of all gene adjacency functions in the corresponding gene module. The approximate scale-free topology can be attained at the soft thresholding power of 30 in the genotypes. (C) Gene modules identified by WGCNA. Gene dendrogram obtained by clustering the dissimilarity based on consensus topological overlap with the corresponding module colors indicated by the color column. Each colored column represents a module, which contains a group of highly connected genes.

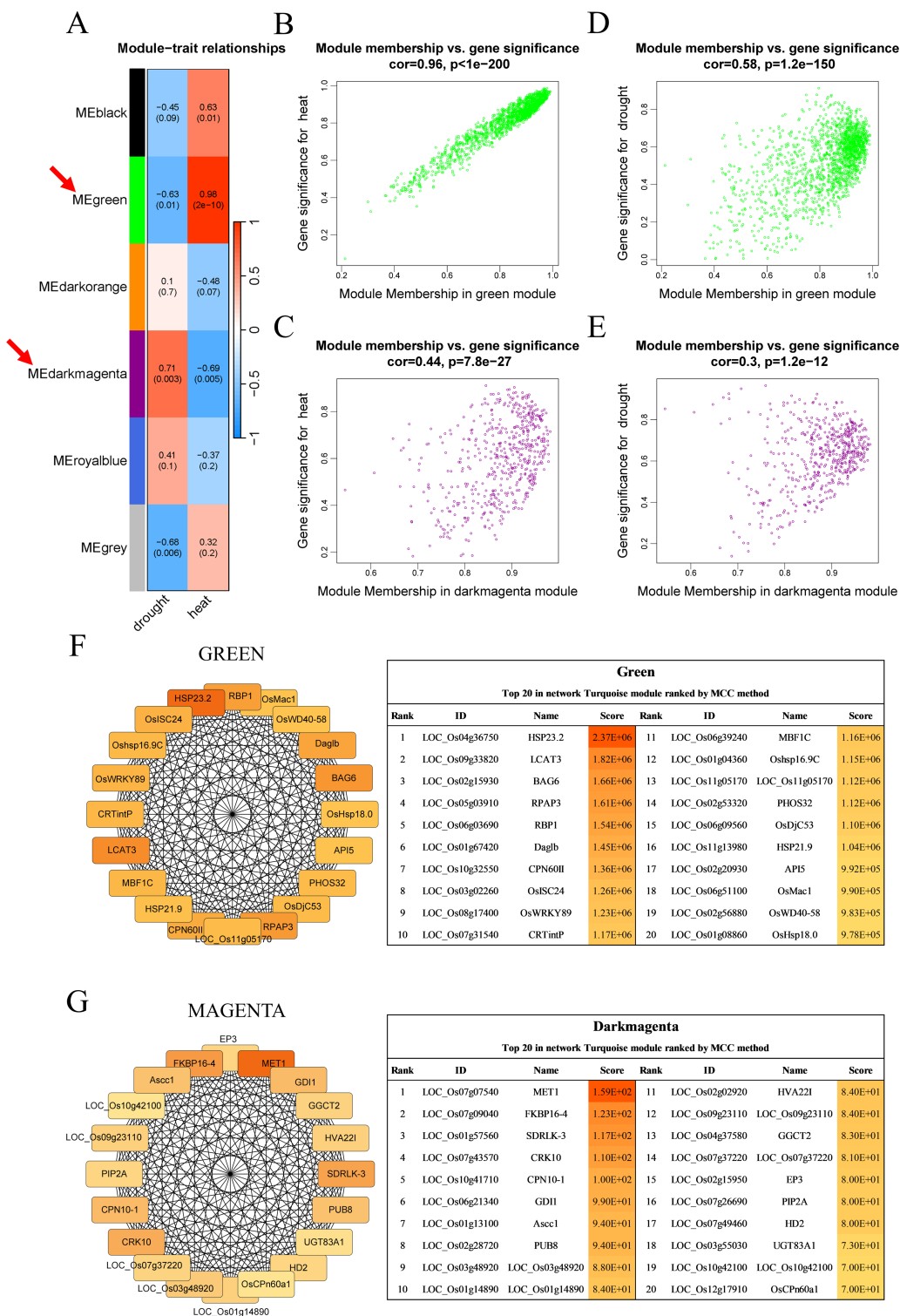

**Figure 3 Co-expression network module analysis.** (A) The correlation coefficient and correlation significance between the module and different stress conditions. Each row in the table corresponds to a consensus module, and each column corresponds to drought or heat stress. Red arrows indicate (continued on next page…)

**Figure 3 (…continued)**
the two modules with the highest correlation. (B–E) Gene significance and module membership fit scatter plot. Each gene is represented by a hollow dot. In (B, D) graphs, the $x$-axis represents the correlation between the module eigengene and the gene expression profile in the green module. The (C, E) graphs correspond to the dark magenta module. In (D, E) graphs, the $y$-axis represents the correlation between the gene and different degrees of drought stress. The (B, C) graphs correspond to heat stress. (F–G) Top 20 hub genes obtained from the interaction network analysis. Identification of hub genes using the maximal clique centrality (MCC) method. Genes with the top 20 MCC values are colored orange to yellow. Orange refers to a relatively large MCC value, whereas yellow refers to relatively smaller MCC values. The F network corresponds to the green module, and the G network corresponds to the dark magenta module.

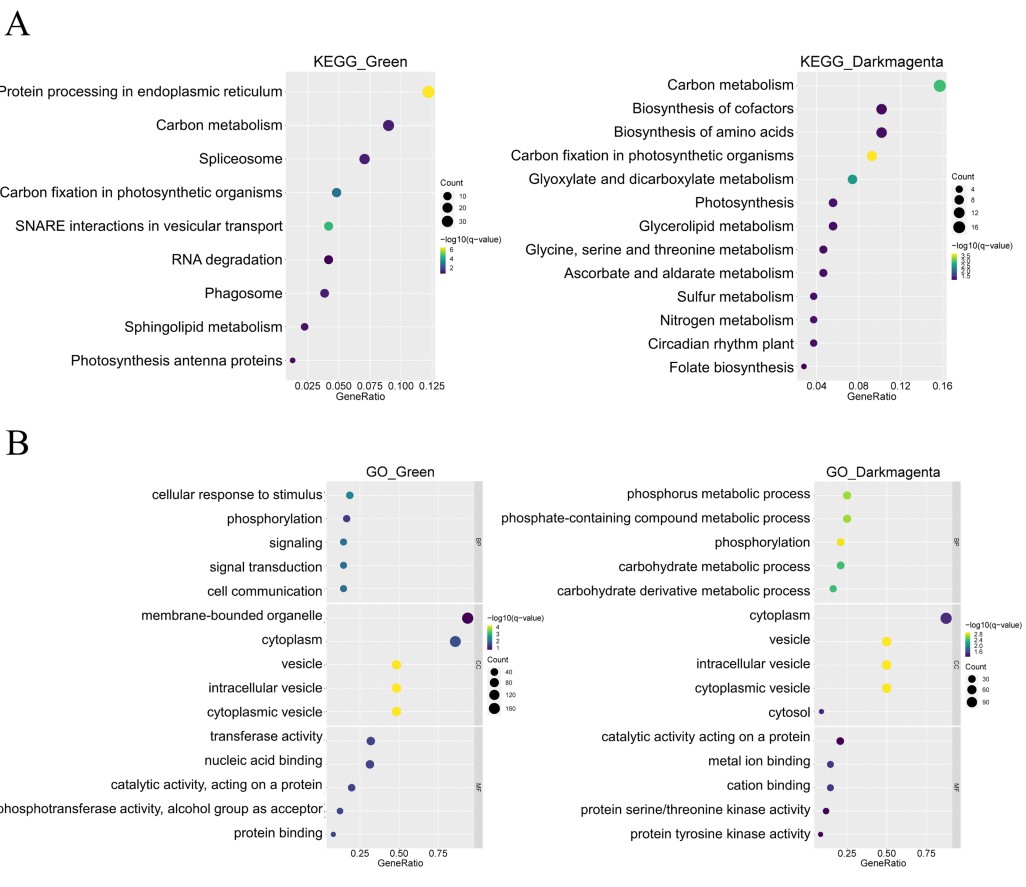

**Figure 4** **Functional enrichment of module genes.** The $y$-axis shows the biological function of a gene in a cell. The $x$-axis represents the ratio of the number of genes enriched from the target pathway to the total genes contained in the gene list. The size of bubble area indicates the number of enriched genes. Bubble color indicates enrichment significance. The green module is shown on the left, and the dark magenta module is shown on the right. (A) Bubble map of the Kyoto Encyclopedia of Genes and Genomes (KEGG) enrichment analysis. (B) Bubble map of the Gene Ontology (GO) enrichment analysis.

group, whereas 1,876 and 3,158 DEGs were upregulated and downregulated, respectively, in the heat stress group.

GO analysis indicated that under drought or heat stress, macromolecule metabolism was the most altered biological process, and hormonal responses and other response activities

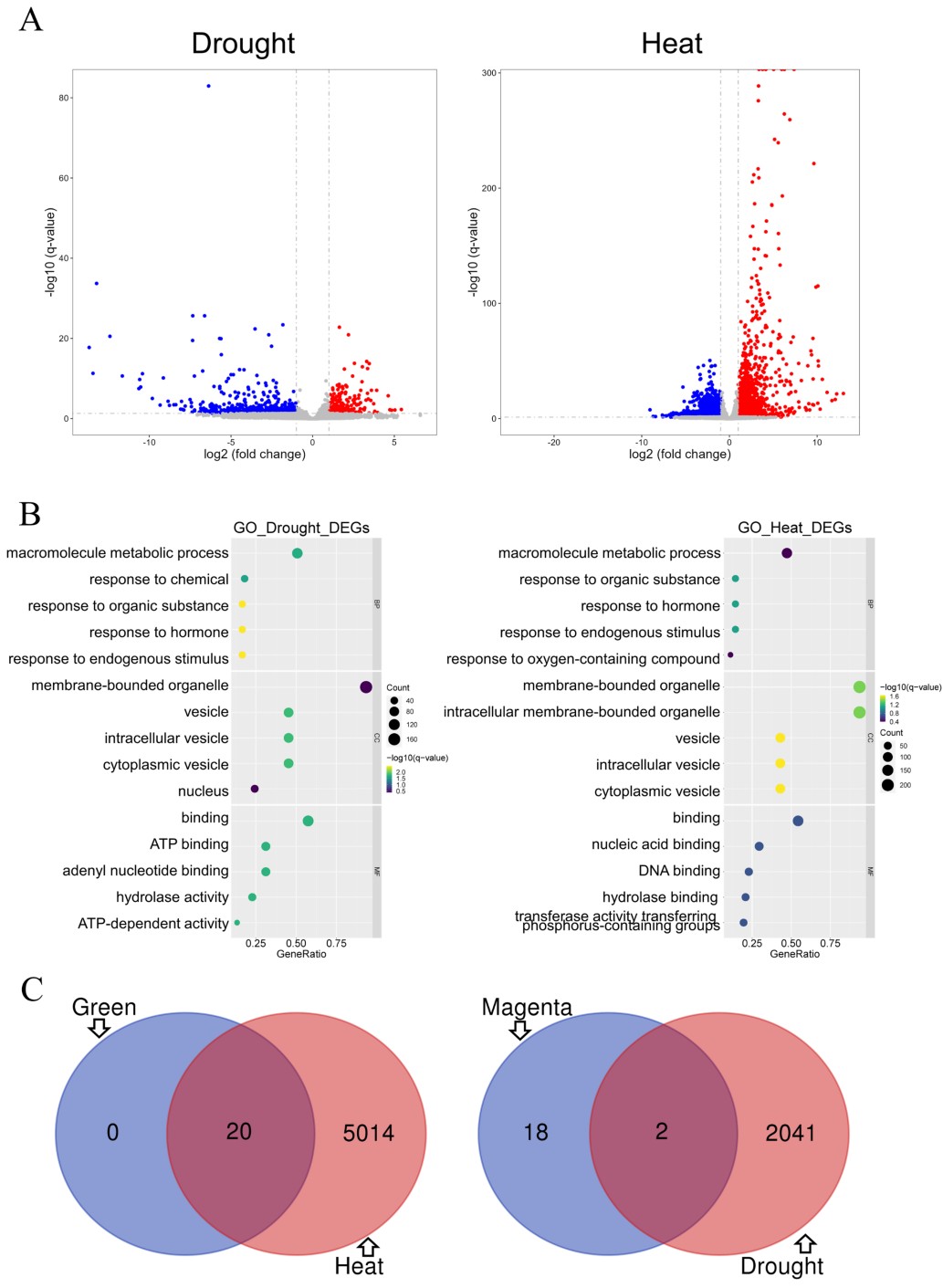

**Figure 5 Analysis of differentially expressed genes.** (A–B) The drought group is shown on the left, and the heat group is shown on the right. (A) Volcano plot of differentially expressed genes (DEGs). Log2 fold change $= 1$ and $p = 0.05$ were used as truncation criteria. The $x$-axis represents log2 fold change, and the $y$-axis represents $-\log10$ $q$-value. Each dot represents a gene. Red dots represent significantly upregulated genes. Blue dots represent significantly downregulated genes. (continued on next page...)

**Figure 5 (…continued)**
Gray dots represent genes with no significant differences. (B) Gene Ontology (GO) enrichment analysis bar chart. The *y*-axis shows the biological function of a gene in a cell. The *x*-axis represents the ratio of the number of genes enriched from the target pathway to the total genes contained in the gene list. The size of bubble area indicates the number of enriched genes. Bubble color indicates enrichment significance. The left panel shows the GO analysis of drought stress and control DEGs, and the right panel shows the GO analysis of heat stress and control DEGs. (C) Venn diagram of intersection of top 20 hub genes and DEGs. Green module and heat stress DEGs are shown on the left. Dark magenta module and drought stress DEGs are shown on the right.

were also altered. The proteins expressed by the two groups of DEGs were mainly binding proteins and mostly located in membrane-bound organelles and vesicles. However, the DEGs in the drought stress group were mostly related to ATP function, whereas those in the heat stress group were involved in nucleic acid and DNA-related functions (Fig. 5B).

The GO enrichment analysis results for the green module and heat stress–induced DEGs shared many similarities. In terms of biological processes, both involved cellular response activities. Regarding molecular functions and cell components, both involve a large proportion of proteins with nucleic acid binding that commonly act on membrane-bound organelles or vesicles. Therefore, the prediction of heat stress-related genes in the green module was expected to be more accurate (Figs. 4B and 5B). DEG analysis in the dataset is an important basis for the subsequent screening of key genes.

## Further screening of hub genes

The top 20 hub genes in the dark magenta and green modules intersected with the DEGs under drought and heat stress, respectively, resulting in the selection of two candidate key genes associated with drought stress and 20 genes associated with heat stress (Fig. 5C).

Green module: The RNA-seq dataset used was GSE168650 (*Kan et al., 2022*), which contained RNA-seq data for two different genotypes of rice subjected to heat treatment and their corresponding controls. The data type was the RAW count. DEGs were analyzed using the same method without setting a threshold filter to identify key genes and their relative expression levels. GEO2R was used to analyze the expression levels of key candidate genes in multiple array datasets, including GSE136746 (*Ps et al., 2017*), GSE41648 (*Sharma et al., 2021*), and GSE14275 (*Hu, Hu & Han, 2009*). Cluster analysis was performed separately according to the original data types. Genes with high expression levels and consistent expression levels among different samples were selected from the heatmaps of both RNA-seq and array data (Figs. 6A–6B, Fig. S1A). Five key genes with the best overall performance were selected: *OsDjC53*, *MBF1C*, *BAG6*, *HSP23.2*, and *HSP21.9*.

Dark magenta module: the dataset GSE159816 (*Zu et al., 2021*) was downloaded, which contained two lines of rice subjected to drought treatment and their corresponding controls. We also analyzed the expression levels of key candidate genes in the GSE93917 (*Wang et al., 2020*) and GSE83378 (*Wei et al., 2017*) array datasets. The results showed that the expression levels of *UGT83A1* and *OsCPn60a1* did not show the same trend in multiple datasets; however, they were classified as key genes for further confirmation (Figs. 6C–6D, Fig. S2B).

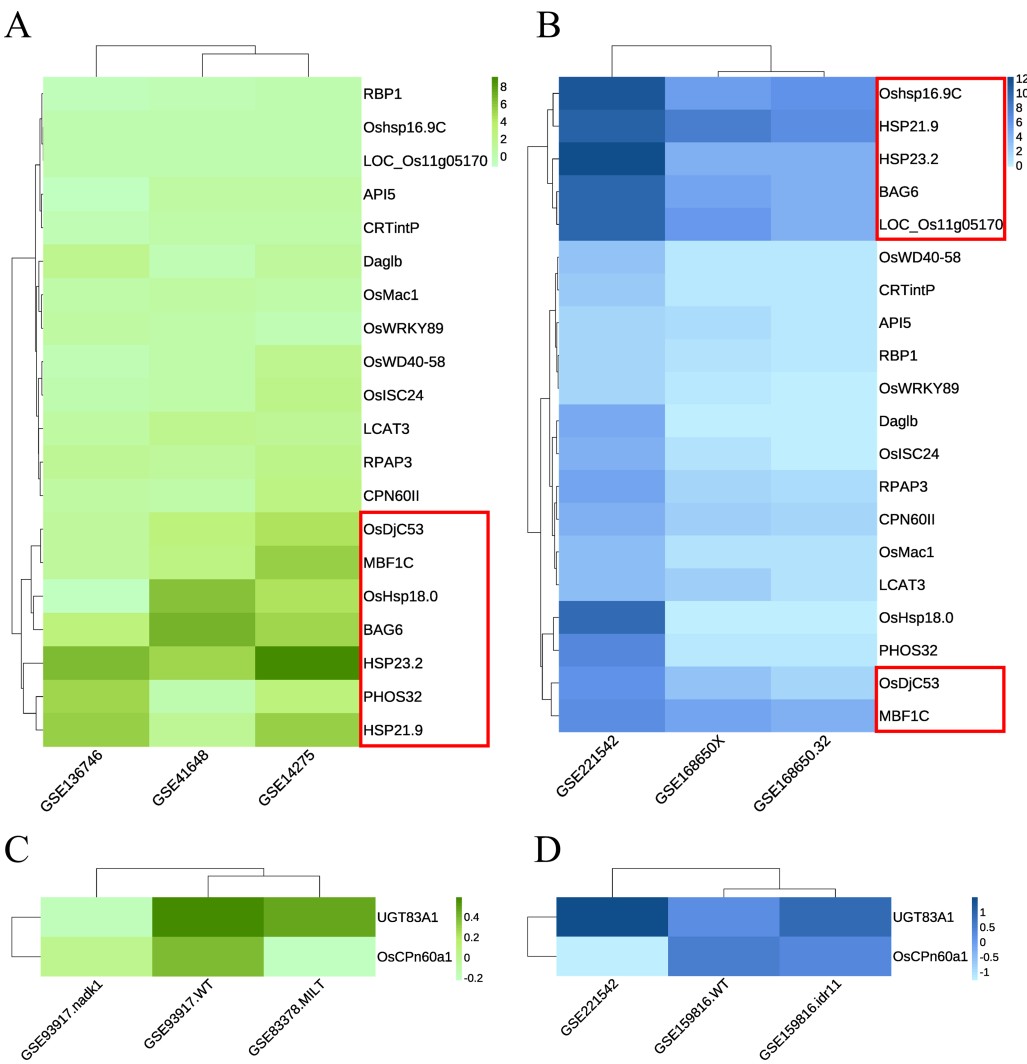

**Figure 6** **Heat maps indicate the expression of candidate key genes in response to heat or drought stress in array sequencing data or RNA-seq data.** The abscissa represents the dataset, and the ordinate represents each candidate key gene. The level of gene expression is indicated by the shade of color. Darker colors indicate a higher expression level. (A, C) Array sequencing data. (B, D) RNA-seq data. (A, B) Heat stress response group. (C, D) Drought stress response group. Red boxes indicate genes with high expression levels across multiple datasets.

## Verification of key genes

qRT-PCR was used to verify changes in the expression levels of key genes in rice subjected to drought or heat stress conditions (Supplementary 2). The results showed that *OsDjC53*, *MBF1C*, *BAG6*, *HSP23.2*, and *HSP21.9* were significantly overexpressed in rice under heat stress conditions (Fig. 7A), whereas the expression levels of *UGT83A1* and *OsCPn60a1* significantly decreased in rice under drought stress conditions (Fig. 7B). In summary, the five candidate genes in the green module may be the key genes associated with the heat stress response in rice.

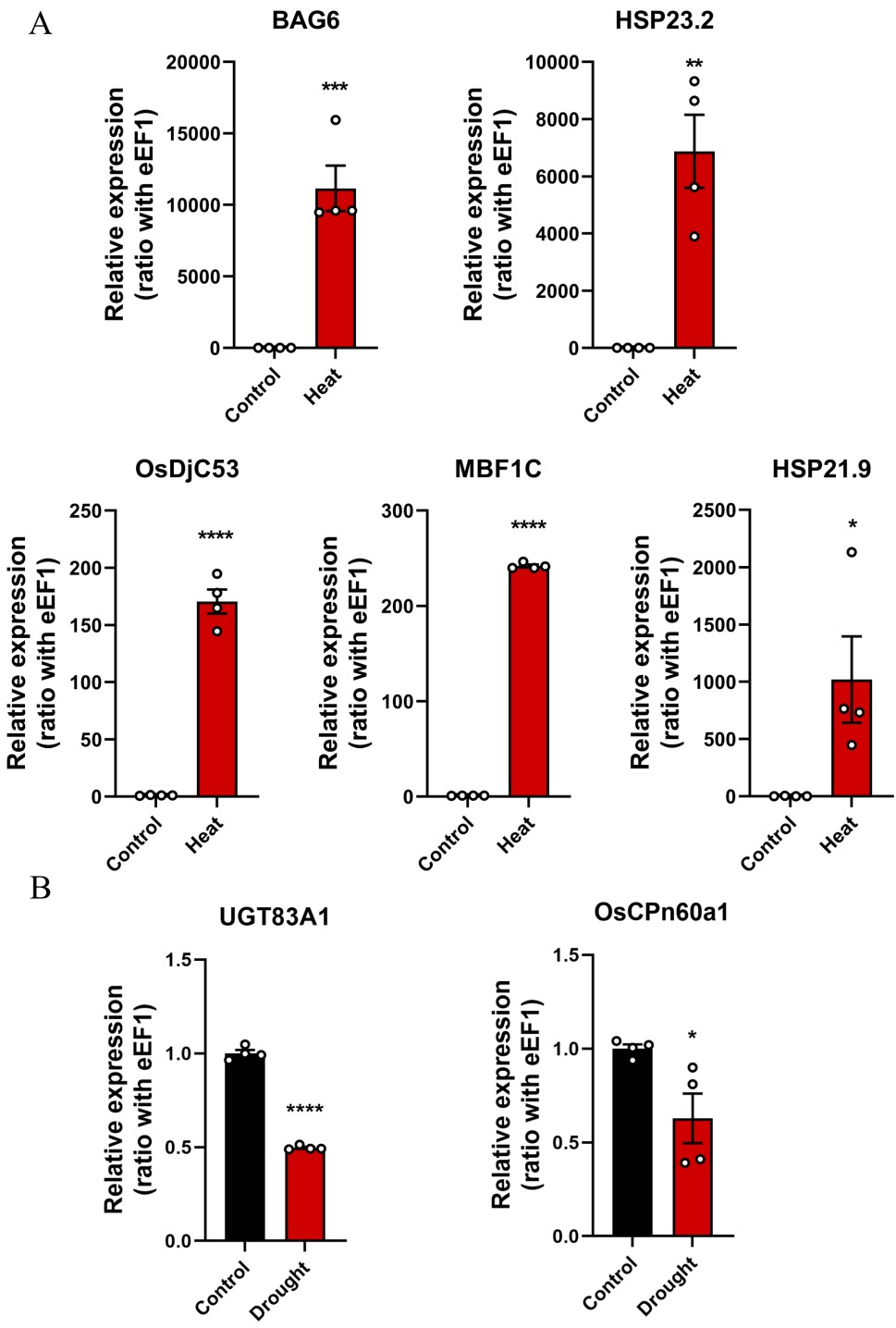

**Figure 7** The expression levels of key genes in rice under drought and heat stress conditions were detected using quantitative real-time PCR (qRT-PCR) and compared with those in the control group. $*p < 0.05$, $**p < 0.01$, $***p < 0.001$, $****p < 0.0001$. (A) Heat stress-related genes, including *BAG6, HSP23.2, OsDjC53, MBF1C,* and *HSP21.9*. (B) Drought stress-related genes, including *UGT83A1* and *OsCPn60a1*.

## DISCUSSION

A co-expression network was constructed using the WGCNA algorithm, which allowed us to identify the top 20 genes and form a core network. Hub genes interact with more genes in the biological regulatory network (*Han et al., 2021*). DEGs represent differences in gene expression levels, indicating their significant roles in stress response (*Han et al., 2021*). To verify the reliability of network analysis results and identify key genes involved in the regulatory network of stress response, the core network was intersected with DEGs identified from the same dataset to obtain candidate key genes associated with the drought and heat stress responses. Furthermore, by analyzing multiple datasets, two key genes responding to drought stress and five key genes responding to heat stress were identified among the candidate key genes. The final qRT-PCR results excluded all key drought-related genes and identified *OsDjC53*, *MBF1C*, *BAG6*, *HSP23.2*, and *HSP21.9* as genes associated with the heat stress response in rice.

UDP-glycosyltransferases (UGTs) are a class of enzymes that add sugars covalently to a wide range of secondary metabolites (*Bowles et al., 2005*). *UGT83A1* is a key gene for yield and drought resistance in rice, and *UGT83A1*- overexpressing lines exhibit strong resistance to drought stress (*Dong et al., 2020*). In addition, the expression level of *UGT85E1* first increases and then decreases under drought stress. The *UGT83A1*- overexpressing line can obviously improve the drought tolerance of rice but is prone to withering (*Liu et al., 2021*). A dataset GSE121303 (*Chung et al., 2016*) was subjected to drought stress for 1–3 days, and *UGT83A1* expression levels fluctuated with drought duration (Fig. S3). In conclusion, the overexpression of *UGT83A1* can improve drought resistance in rice; however, *UGT83A1* expression levels do not necessarily increase or decrease when rice is under drought stress. This suggests that *UGT83A1* may be involved in the drought stress response through a complex mechanism influenced by other factors.

*OsCPn60a1* may bind to the RuBisCO small and large subunits and is implicated in the assembly of the enzyme oligomer (*Aigner et al., 2017*). Thus, we suggest that changes in *OsCPn60a1* expression levels may indicate changes in photosynthesis but may not necessarily be directly associated with the drought stress response.

The response of rice to heat stress is closely linked to HSPs. There is a high degree of homology between HSP21.9 and HSP23.2 proteins (Fig. S4). Furthermore, protein motif prediction revealed multiple shared motifs among OsDjC53, MBF1C, BAG6, HSP23.2, and HSP21.9 (Fig. S5), indicating that these five proteins potentially interact or cooperate with each other. Interestingly, we reproduced the stress treatment used in the dataset (GSE221542), and). heat stress did not result in visible changes (Fig. S6).

HSPs are crucial for plant growth and abiotic stress tolerance (*Mansfield & Key, 1987*; *Sarkar, Kim & Grover, 2009*). *OsDjC53* is predicted to belong to the *DnaJ/HSP40* family (RGI). *HSP21.9* and *HSP23.2* belong to the HSP20 family (RGI). HSPs were found to control programmed cell death of suspension cells in response to high temperatures and play an important role in the response to hyperosmotic and heat shock stress by preventing the aggregation of stress-denatured proteins and by disaggregating proteins (*Wang et al., 2019*). *MBF1C* is a multi-protein bridging factor. In *Arabidopsis*, *MBF1C* improves the

tolerance to heat and osmotic stress by partially activating or disrupting the ethylene response signal transduction pathway (*Suzuki et al., 2005*). *Bcl-2*-associated athanogenes (*BAGs*) are considered to be adaptors that can form complexes with signaling molecules and molecular chaperones (*Kabbage & Dickman, 2008*). *BAG6* plays a critical role in plant heat tolerance by regulating the accumulation of HSPs and maintaining protein homeostasis under heat stress conditions in *Arabidopsis* (*Echevarría-Zomeño et al., 2016*). In other species, these genes are also highly correlated with drought resistance. Thus, increased expression of these genes may improve the ability of rice to resist heat stimulation.

This study has a few limitations. The small number of controlcontrols, heat stress, and drought stress samples in the WGCNA may have resulted in potential statistical errors during the construction of the co-expression network. The available data in the GEO validation queue is extremely limited, which restricts the validation of key genes across a wider range of stress durations and intensities, thus preventing further analysis of their utility and stability. This underscores the need for more publicly available transcriptomic sequencing data. Furthermore, additionalFurthermore, a experiments are required to elucidate the mechanisms underlying the response of rice to drought and heat stress. AlthoughA our analytical method successfully predicted the heat stress response genes in rice, it did not perform as well in predicting drought stress response genes, possibly because of the limited data used by the WGCNA. We found low-quality data in the drought group (Fig. S7). Therefore, higher-quality and larger datasets are required for more accurate analyses and predictions. Future studies will need more data to find key genes.

In rice stress response research, we hope that more transcriptome data of different subspecies, tissue types, growth stages in different stress types, duration gradients, and intensity gradients can be published. This is because machine learning and artificial intelligence will be able to predict key genes more accurately in future research, but they require extremely large data (*Xu et al., 2023*).

Overall, our findings provide valuable insights into the molecular mechanisms underlying the response of rice to drought and heat stress and may have important implications for the development of stress-tolerant rice varieties through genetic engineering approaches.

## CONCLUSIONS

Our approach successfully identified key candidate genes associated with heat stress response in rice. More importantly, our study represents an innovative integration of multiple RNA-seq and array datasets from the GEO database to analyze the key genes associated with drought and the heat stress responses in rice. The degree of fit between each module and the corresponding trait (Fig. 3B and 3E) determined the effectiveness of the obtained key genes (Fig. 7).

## ACKNOWLEDGEMENTS

The authors acknowledge the assistance of Jun Hu from College of Life Science, Wuhan University, China, who offered great help in providing important technical guidance and rice seeds.

### Funding

This research was funded by the Open Research Fund of State Key Laboratory of Hybrid Rice (Wuhan University, China), grant number KF202202; China Postdoctoral Science Foundation (2020TQ0363 and 2020M682598); Natural Science Foundation of Hunan Province: 2021JJ40992; the National Natural Science Foundation of China (82300787); and the Fundamental Research Funds for the Central Universities of Central South University (2023ZZTS0571 and 2023ZZTS0995). The funders had no role in study design, data collection and analysis, decision to publish, or preparation of the manuscript.

### Grant Disclosures

The following grant information was disclosed by the authors:
The Open Research Fund of State Key Laboratory of Hybrid Rice (Wuhan University, China): KF202202.
China Postdoctoral Science Foundation: 2020TQ0363, 2020M682598.
Natural Science Foundation of Hunan Province: 2021JJ40992.
The National Natural Science Foundation of China: 82300787.
The Fundamental Research Funds for the Central Universities of Central South University: 2023ZZTS0571 and 2023ZZTS0995.

### Competing Interests

The authors declare there are no competing interests.

### Author Contributions

- Gaohui Cao conceived and designed the experiments, performed the experiments, analyzed the data, prepared figures and/or tables, authored or reviewed drafts of the article, and approved the final draft.
- Hao Huang conceived and designed the experiments, performed the experiments, analyzed the data, prepared figures and/or tables, authored or reviewed drafts of the article, and approved the final draft.
- Yuejiao Yang performed the experiments, authored or reviewed drafts of the article, and approved the final draft.
- Bin Xie performed the experiments, authored or reviewed drafts of the article, and approved the final draft.
- Lulu Tang conceived and designed the experiments, performed the experiments, authored or reviewed drafts of the article, and approved the final draft.

## Data Availability

The raw data are available in the Supplemental File.

## Supplemental Information

Supplemental information for this article can be found online at http://dx.doi.org/10.7717/peerj.17255#supplemental-information.

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
