# Peer review of "Analysis of drought and heat stress response genes in rice using co-expression network and differentially expressed gene analyses"

_PeerJ, doi:10.7717/peerj.17255_

## Round 0.1 · original submission · Major Revisions

Climate change including drought and heat stress is a major issue for rice farming. This study does provide some significant insights into the molecular mechanisms of drought and heat stress using available data from the public domain.
Authors are requested to follow and answer all the valuable comments from all 4 reviewers including the following major comments:

1. Methodology: Why after sterilization seeds were soaked at 37 °C? Explain in detail the treatment for drought and heat stress followed in this study.

Heat and drought stress: Why are only 1hr and at 37 °C and 12 h period used for drought treatment? any rationale behind it?

Also include plant pics in control and treatment group in results section.

2. WGCNA Network Construction:
Please elaborate on the rationale for selecting the beta value in the adjacency matrix construction. Also, it would be helpful for the researchers if you could explain how the chosen beta value contributes to the creation of a scale-free network.

3. Normalization of Gene Expression:
Kindly, give the details of the specific normalization techniques applied to gene expression levels and their impact on downstream analyses as clarity on normalization methods adds transparency to the analysis process.

4. Intersection of Hub and DEGs:
Please discuss the biological significance of the intersection between the top 20 hub genes and the DEGs and elaborate on how this intersection contributes to the identification of candidate key genes.

5. Validation Cohorts and Data Availability:
The limited availability of data in the GEO validation cohorts is highlighted. It would be beneficial if you can discuss the implications of this limitation on the overall robustness of the findings. Additionally, providing recommendations for future studies to enhance the availability of transcriptome sequencing data would strengthen the research.

6. Prediction Performance and Dataset Size:
The study acknowledges variations in prediction performance between heat and drought stress response genes. Discussing the reasons for this discrepancy and proposing potential strategies for improving prediction accuracy, such as acquiring larger datasets for WGCNA analysis, would enhance the study's depth.
Mechanistic Insights:
While the study successfully identifies key candidate genes, more mechanistic insights into how these genes function in the context of drought and heat stress responses could be beneficial. Providing a brief discussion or proposing future experimental directions to unravel these mechanisms would strengthen the implications of the findings.

7. Clarity on Heat Stress Response Genes:
The study mentions the successful prediction of heat stress response genes but suggests limitations in predicting drought stress response genes. Kindly clarify the specific challenges faced in predicting drought stress response genes and discuss potential avenues for overcoming these challenges.


Addressing these points will enhance the clarity and completeness of the methodology section, contributing to a more comprehensive understanding of the research approach. Further, these points will enhance the clarity, depth, and potential impact of the study's findings. Overall, the study presents valuable insights, and these recommendations aim to further strengthen its scientific rigor and contribution to the field.

Reviewer 1 ·

Basic reporting

English Language: This paper have good written in English ,clear, and technically correct text.
Literature references: The article have sufficient introduction and background to demonstrate related abiotic stress in rice. All references site appropriately.

structure, figures, tables. Raw data shared: The figure, table and raw data are share nicely. All figure and table have sufficient resolution, and appropriately described and labeled and raw data have been made available

Self-contained with relevant results to hypotheses : The hypothesis of this articles is not more fitted. The validation of the gene expression done by qRT-PCR and using few primers.

Experimental design

This is not primary research. This work mostly previous published data are used except of qRT-PCR.
Research question well defined, relevant & meaningful: The article is not clearly define the research question but presenting data will be useful for readers.
It is stated how research fills an identified knowledge gap : This article try to fill up the gap of drought and heat stress response genes in rice by co-expression network and differentially expressed gene analyses using previous published papers.
Rigorous investigation performed to a high technical & ethical standard: The investigation not up to rigorously and to a high technical standard. Bioinformatics tools are more prominent in this paper compare to validation
Methods described with sufficient detail & information to replicate: Methods are well described in this paper

Validity of the findings

Impact and novelty not assessed. There is not novel finding are presented in this paper. However, useful for readers.
All underlying data have been provided; they are robust, statistically sound, & controlled: All underlying data have been provided in this paper is acceptable for this journal. All the data are well analyzed by standard statistical tools
Conclusions are well stated, linked to original research question & limited to supporting results: Conclusion is very brief and forcing of fulfilling the hypothesis

Additional comments

No

·

Basic reporting

The English and flow of the manuscript is ok. Citation and references are appropriate.

Experimental design

The data is gathered from publicly available RNAseq data, not generated by the authors. But the research questions, goal and flow are well defined. However, authors can try reproducing this data to check for the same results.

Validity of the findings

Authors can comment on the heat and drought application in the original studies who submitted the data in GEO. Gene expression might vary due to type, duration of stages of heat application. Authors need to clarify this issue with appropriate justification.

Additional comments

1. WGCNA has another elaboration weighted gene co-expression network analysis.
2. In some places scientific names are not italicized
3. Naming the groups as color is not acceptable since it might not be detectable to a colorblind person or in black and white printing.
4. In some sections of the result, description appropriate for methodology has been included.

Reviewer 3 ·

Basic reporting

1. In lines 48 to 57, please list which of these datasets are RNA-seq and which of them are microarray data.
2. In Line 57, why authors only introduce GSE221542 in detail and do not introduce other datasets?
3. What is "MCC" in line 70? The abbr. should be spelled out for the first occurrence.
4. Please provide R scripts to improve the reproducibility.
5. There are 3 water levels in dataset GSE221542. How do authors select the samples as the case-control samples for differential gene analysis?

Experimental design

1. The author does not analyze the batch effect of the dataset. If there is a batch effect in the dataset, take GSE221542 as an example, the result of clustering or differential gene analysis may be highly relevant to the batch rather than the heat or drought.
2. Please analyze the consistency of the gene expression among different datasets including RNA-seq data and microarray datasets. If the material and experiment are similar, the gene expression should be consistent. Otherwise, some low-quality datasets should be discarded.
3. Why were the top 20 genes selected in the WGCNA analysis? Please provide suitable reasons or statistics.

Validity of the findings

Please add the batch effect analysis before data processing. If there are batch effects, please remove batch effects or remove some low-quality samples.

Reviewer 4 ·

Basic reporting

The manuscript entitled "Analysis of drought and heat stress response genes in rice using co-expression network and differentially expressed gene analyses" by Gaohui Cao and colleagues investigates the genetic basis of rice's response to drought and heat stress. The study integrates multiple datasets from the GEO database and employs the WGCNA algorithm to construct a co-expression network. The authors identified key genes associated with heat stress response (OsDjC53, MBF1C, BAG6, HSP23.2, and HSP21.9) and potentially affected by drought stress (UGT83A1 and OsCPn60a1). These findings were verified using quantitative real-time PCR.
Overall, the manuscript is logically structured, well-organized, and the findings are supported by robust statistical analysis. The use of advanced analytical techniques and the integration of diverse datasets provide a comprehensive view of the stress response in rice. The context and hypothesis are clearly outlined, and the study's results offer significant insights. Moreover, the authors discussed the biological applications and current research on the identified key genes.

Experimental design

Well designed.

Validity of the findings

1. Please improve the resolution of Figure 4 and Figure 5B.
2. Figure 5C, it’s better to add the label for each color block which represents top 20 hub genes and DEGs respectively.

Additional comments

1. Line 251, suggest “heat stress response” to be “the heat stress response”. Line 254, suggest “key gene” to be “key genes”.
2. Suggest checking the references throughout the whole manuscript, when clicking on the cited references within the article, the hyperlinks do not navigate to the references section. Also, many references cited in the manuscript have space in between the previous text while some don’t, and some citations appear before the period while others are placed afterwards.

---

## Round 0.2 · accepted · Accept

This revised version looks much better, and authors corrected and included the missing points as reported earlier.


Reviewer 1 ·

Basic reporting

Authors make correction at all basic reporting unites such as Clear and unambiguous, Literature and reference table and figure structure and contents.

Experimental design

I was all ready point outed experimental designing of this manuscript is good and improved after corrections

Validity of the findings

Validation and novelty are expressing as majorly dry lab system so that this work may permissible as novelty

Additional comments

There is no any additional comments may accept for publication

Reviewer 3 ·

Basic reporting

The problems have been improved.

Experimental design

The problems have been improved.

Validity of the findings

The problems have been improved.

Reviewer 4 ·

Basic reporting

The authors improved the quality of figures and revised the manuscript text for better clarity. Moreover, they addressed and corrected several identified mistakes and inconsistencies. Based on revision of manuscript, authors successfully incorporated the point to point wise through the manuscript as suggested comments so that I would like to strongly recommend that acceptance of manuscript in your prestigious journal for publication.

Experimental design

NA

Validity of the findings

NA

Additional comments

NA